# Fluorescent Protein Inserts in between NC and SP2 Are Tolerated for Assembly, Release and Maturation of HIV with Limited Infectivity

**DOI:** 10.3390/v11110973

**Published:** 2019-10-23

**Authors:** Mourad Bendjennat, Saveez Saffarian

**Affiliations:** 1Radiation Oncology Department, University of Miami Miller SOM-SCCC, Miami, FL 33136, USA; 2Department of Physics and Astronomy, University of Utah, Salt Lake City, UT 84112, USA; 3Center for Cell and Genome Science, University of Utah, Salt Lake City, UT 84112, USA; 4Department of Biology, University of Utah, Salt Lake City, UT 84112, USA

**Keywords:** fluorescent HIV, NL4-3(dGFP), R8(dGFP), NL4-3(iGFP)

## Abstract

We report the design of a fluorescent HIV construct that is labeled by insertion of fluorescent protein between the nucleocapsid (NC) and spacer peptide 2 (SP2) domains of Gag and further show that the fluorescent protein is released from its confines within Gag during maturation. This fluorescent HIV is capable of budding and maturation with similar efficiency to the parental virus. Virions generated using this design within the R8 HIV backbone pseudotyped with VSV-G were capable of delivering small RNA genomes encoding GFP to the target cells; however, the same design within the NL4-3 backbone has limited HIV infectivity. The virions generated by these constructs are approximately 165 ± 35 nm in size, which is significantly larger than wild type HIV. We suggest that this design has the potential to be a vehicle for protein and small guide RNA delivery.

## 1. Introduction

Fundamental research into mechanism of HIV virus replication has uncovered a complex release and maturation machinery that leads to delivery of RNA genomes to host cells [1]. This process has been successfully utilized in the design of lentiviral vectors [2].

Individual HIV virions assemble on the plasma membrane of cells incorporating ~2000 Gag and ~120 Gag–Pol proteins, along with HIV accessory proteins and two copies of genomic viral RNA [1]. During assembly, Gag and Gag–Pol proteins form the immature HIV lattice on the inner leaflet of the viral membrane [3]. The immature virions at the final stages of their assembly recruit Endosomal Sorting Complexes Required for Transport proteins to catalyze their release from the plasma membrane [4,5,6,7,8]. Once released, the immature viral lattice is subjected to specific processing by HIV protease to produce the mature infectious virion [9,10]. Efficient release of virions capable of maturation requires timely release of the immature virions before activation of the protease. An untimely activation before release can lead to release of non-infectious virions void of Pol associated enzymes [11]. Gag is composed of MA, CA, SP1, NC, and SP2 domains followed by p6. During HIV virion assembly, both Gag and Gag–Pol attach to the virion membrane’s inner leaflet through the myristylated N-terminus of the MA domain [12,13,14]. The Gag CA domain harbors most of the interfaces required for formation of both immature as well as mature HIV capsids [15,16,17,18,19,20,21]. The NC domain has a primary function in recruiting and packaging genomic RNA [22,23,24] and is also reported to interact with cellular factors during assembly [25]. Complex coordination between many of the viral domains as well as cellular protein interactions are required for efficient release; therefore, the architecture of Gag and Gag–Pol domains are evolutionary optimized to support maximum efficiency for release of infectious virions.

Insertion of foreign proteins within the architecture of Gag is, however, poorly tolerated. The first attempt at creating a fluorescent HIV virion was carried out by inserting a fluorescent protein fused to MA [26], which was later modified to incorporate a cleavage site in between MA and the fluorescent protein [27]; both of these designs only partially support full virion infectivity.

In this manuscript we set out to find additional insertion sites that are tolerated for efficient HIV virion release and maturation.

## 2. Materials and Methods

### 2.1. Expression Vectors, Cells, and Antibodies

HIV-1 *∆*R8.2 (HIV-1_NL4-3_ R9 *∆NEF. ∆ENV* [28]) was used.

All cell lines used were grown in complete DMEM medium under standard conditions, except during TIR-FM experiments, where cells were incubated in CO_2_-independent medium (LifeTechnologies, Carlsbad, CA, USA).

Anti-p24 (183-H12-5C, NIH AIDS Reagent Program, Germantown, MD, USA), anti-GFP (sc-8334, Santa Cruz Biotech, Dallas, TX, USA), anti-mCherry (TA150125, Origene, Rockville, MD, USA), anti-RT (MAb21, NIH AIDS Reagent Program), and infrared dye coupled secondary antibodies (LI-COR, Lincoln, NE, USA) were used for immunoprobing. Scanning was performed with the Odyssey infrared imaging system (LI-COR) in accordance with the manufacturer’s instructions at 700 and/or 800 nm, accordingly.

### 2.2. Construction of the Fluo-R8.2 Battery

According to the five major processing sites previously characterized [24], we inserted in frame the fluorescent proteins (Fluo-proteins) in Gag ORF of R8.2, conserving the sequence of each individual site intact, although duplicating them for flanking Fluo-proteins at N- and C-terminuses, accordingly.

*Gag sequence:* MA|CA|SP1|NC|SP2|p6     [|: PR cleavage sites]MA–*Fluo*-CA: MA....SQNY|PIV–*Fluo*-SQNY|PIV....CACA–*Fluo*-SP1: CA....ARVL|AEA–*Fluo*-ARVL|AEA....SP1SP1–*Fluo*-NC: SP1....ATIM|MQR–*Fluo*-ATIM|MQR....NCNC–*Fluo*-SP2: NC....RQAN|FLGEF–*Fluo*-RQAN|FLGEF....SP2

The Fluo-R8.2–STOP constructs were generated by introducing a translation stop codon immediately after the Gag p6 domain.

### 2.3. Virion Release Analysis

The 293T cells were transfected accordingly using the standard CaPO_4_ precipitation technique. Both cells and media were collected for analysis. Cells were lysed in RIPA buffer (140 mM NaCl, 8 mM Na_2_HPO_4_, 2 mM NaH_2_PO_4_, 1% NP-40, 0.5% sodium deoxycholate, 0.05% SDS). After removal of residual cell debris by centrifugation, virions were pelleted from cell supernatants by centrifugation for 2 h through a 10% (*w/v*) sucrose cushion at 15,000× *g*. Virion pellets were re-suspended in PBS. Both cells and virions were analyzed by SDS–PAGE and immunoblotting. Band intensities were quantified using the LI-COR Image Studio Line software. Virion release yields/ratio were calculated as virion-associated Gag/Gag–Pol forms per cell-associated Gag/Gag–Pol forms based on CA probing.

### 2.4. TIR-FM Assessments

HeLa cells were transfected using Lipofectamine 2000 (LifeTechnologies Carlsbad, CA, USA). Live images were acquired using an iMIC digital microscope made by TILL photonics controlled by TILL’s Live Acquisition imaging software. The TIRF critical angle was verified by scanning the laser beam across the back aperture and measuring the reflection of the laser from the glass sample interface back into the objective and onto the quadrant photodiode.

### 2.5. Efficiency of RNA Packaging and Delivery

Virions were produced using 293T cells grown in 6 cm dishes. Cells were co-transfected following the standard CaPO_4_ precipitation technique with either parental R8.2 (non modified) or R8.2:Gag–Fluo constructs along with pLOX–GFP [29] and pCMV–VSV–G; then, media were replaced 4 h post-transfection with fresh ones. Then 32 h later, supernatants were harvested and syringe-filtered through 0.45 µm membranes. Viral titers were estimated using fluorescence-activated cell sorting (FACS) to detect eGFP expression that is driven by the packaged pLOX–GFP mRNAs and transduced in the infected HeLa cells. The infectivity values are relative to the parental R8.2 vector.

### 2.6. Infectivity

The supernatant of 293T cells was collected 48 h after transfection with viral vectors and then added to a monolayer of TZM-b1 cells. TZM-b1 cells were then harvested using the Britelite Plus Reporter Gene Assay (Perkin Elmer, Waltham, MA, USA). The infectivity was quantified by reading luminescence using the Cytation 5 microscope (Fisher Scientific Company, LLC. Hanover Park, IL, USA).

### 2.7. Electron Microscopy

HeLa cells were grown on ACLAR disks and transfected with either NL4-3 or NL4-3(NC–Fluo-SP2) vectors. Cells were fixed in 2.5% glutaraldehyde plus 1% paraformaldehyde in 0.1 M cacodylic buffer for 30 min and then embedded in resin using an Embed 812 kit (Electron Microscopy Sciences, Hatfield, PA, USA) and sectioned at 80 nm with a diamond knife (Diatome) using a Leica EM UC6 (Leica Microsystems, Wetzlar, Germany). Sections were visualized using a JEM 1400 Plus electron microscope (JEOL, Tokyo, Japan) at 120 kV.

## 3. Results

### 3.1. Insertion of Fluorescent Proteins between NC and SP2 Is Tolerated by HIV for Virion Release and Maturation

As a first test for functional release of fluorescent HIV vector that incorporates fluorescent proteins within the open reading frame of Gag, we used the HIV ∆R8.2 (R8.2) as our backbone. R8.2 is derived from the full length HIV-1 R9 vector and incorporates all components of R9 except ENV and NEF [28]. HIV Gag consists of MA, CA, SP1, NC, SP2, and p6 domains. Among other trials, we placed the fluorescent protein open reading frame (ORF) at the junction between Gag specific domains, resulting in production of four R8.2:Gag–Fluo vectors: R8.2:Gag(MA–Fluo-CA), R8.2:Gag(CA–Fluo-SP1), R8.2:Gag(SP1–Fluo-NC), and R8.2:Gag(NC–Fluo-SP2) (Figure 1A).

R8.2:Gag–Fluo vectors were then tested following the virion release assay using 293T cells and harvesting viruses from cell supernatants 24 h post-transfection. Immunoblots depicted in Figure 1B show similar protein expression levels from the four R8.2:Gag–Fluo constructs with either active or inactive protease accordingly; however, in comparison to the R8.2 parental virus (WT), only the R8.2:Gag(NC–Fluo-SP2) vector produced a similar yield of virions released and Gag/Gag–Pol processing as assessed using CA (p24) probing. All other R8.2:Gag–Fluo vectors showed defects in virion release and an alteration of the Gag/Gag–Pol maturation profile as visualized by the CA/Gag precursor ratio in purified virions.

We further assayed the virion assembly using TIR-FM imaging of transfected HeLa cells 12 h post-transfection. As shown in Figure 1C, all three R8.2:Gag–Fluo vectors aside from R8.2:Gag(NC–Fluo-SP2) show low densities of formed virions at the plasma membrane of R8.2:Gag–Fluo expressing cells, which suggested the defect within other three vectors as likely related to nucleation and assembly of virions on the cell surface. Insertion of other fluorescent proteins including mCherry, eGFP, dendra2 and mTagBFP2 in R8.2:Gag(NC-Fluo-SP2) did not affect the overall efficiency of virus budding, maturation and infectivity (Appendix A).

### 3.2. Defects in Virion Assembly and Release When Fluorescent Proteins Are Inserted between MA and NC

To further test our hypothesis that the insertion of Fluo-proteins between Gag MA and NC domains affect negatively the assembly process and therefore the yield of virions produced, we inserted in the R8.2:Gag–Fluo vectors a stop codon immediately after the Gag p6 domain (R8.2:Gag–Fluo-STOP) and analyzed the kinetics of virion release by transfected 293T cells (Figure 2). R8.2:Gag–Fluo-STOP vectors expressed only Gag and no Gag–Pol to avoid its potential additional interference during the Gag assembly process. Our results clearly showed a delay in virion release when Fluo-proteins were inserted between CA and NC domains (CA–Fluo-SP1–STOP and SP1–Fluo-NC–STOP) and to a lesser extent, although still substantial, when placed between MA and CA domains (MA–Fluo-CA–STOP), while Fluo-protein insertion between NC and SP2 (NC–Fluo-SP2–STOP) had a minimal effect when compared to the parental unlabeled Gag (WT–STOP). Interestingly, no significant interference of Gag–Pol expression and loading in assembling virions was observed when Fluo-proteins were inserted between NC and SP2 based on the similar time course and yield of virion release (WT versus NC–Fluo-SP2). To this end, for a more accurate comparative analysis between all constructs involved during this specific analysis, we inactivated the HIV protease in the constructs with no stop codon inserted after the Gag p6 domain for the stabilization of incorporated Gag–Pol during virion assembly and release.

### 3.3. Virions with Insertion of Fluorescent Proteins between NC and SP2 Can Efficiently Carry the GFP Gene to the Host Cell

For assessment of R8.2:Gag–Fluo constructs, virions were produced using 293T cells as described in the Materials and Methods section. Equal load of virions, based on density of the p24 band, were used for western blot analysis as shown in (Figure 3A). Interestingly, a clear defect is seen in the virion Gag/Gag–Pol maturation profile of R8.2:Gag(MA–Fluo-CA), R8.2:Gag(CA–Fluo-SP1), and R8.2:Gag(SP1–Fluo-NC) as noticed by the CA(p24)/Gag–Fluo ratios when compared to R8.2 (WT; p24/Gag) and R8.2:Gag(NC–Fluo-SP2; p24/Gag–Fluo). To quantify the genome delivery of various constructs, equal load of virions, based on the density of the p24 band as shown in Figure 3A, were used to infect HeLa cells. Figure 3B shows the Hela cells 48 h after infection. Fluorescence microscopic evaluation clearly indicated that the parental R8.2 (WT) and R8.2:Gag (NC–Fluo-SP2) could deliver the GFP gene more efficiently compared to the rest of the R8.2:Gag–Fluo constructs. Quantification of the GFP levels of R8.2:Gag–Fluo virions indicated that VSVG pseudotyped R8.2:Gag (NC–Fluo-SP2) virions were ~90% infectious compared to VSVG pseudotyped parental R8.2 virions. The VSVG pseudotyped R8.2:Gag (CA–Fluo-SP1) and VSVG pseudotyped R8.2:Gag (SP1–Fluo-NC) were non-infectious, and VSVG pseudotyped R8.2:Gag (MA–Fluo-CA) was ~18% infectious compared to the VSVG pseudotyped parental R8.2 (Figure 3C).

### 3.4. Insertion of Fluorescent Protein between NC and SP2 within NL4-3 Does Not Support Full Infectivity of Virions

To test the infectivity of the NC–Fluo-SP2 virions in the context of packaging full HIV genome, we generated the NL4-3 (NC–Fluo-SP2) backbone. As shown in Figure 4A, the insertion of NC–Fluo-SP2 supports efficient proteolysis of Gag and release of the capsid proteins. The released virions however were not nearly as infectious as WT NL4-3 as shown in Figure 4C. Further analysis of the NL4-3 (NC–Fluo-SP2) virions revealed highly heterogeneous sizes with an average of 160 ± 35 nm compared to 130 ± 15 nm measured for WT virions. A representative EM image of virions released from HeLa cells plated and transfected on ACLAR is shown in Figure 4C.

## 4. Discussion

Fundamental research in packaging and genome delivery of HIV had over the past 20 years led to development of potent life technology tools including the lentiviral delivery systems that are now commonly used. These lentiviral delivery systems rely on Gag and Gag–Pol proteins, which in unison drive viral assembly release and maturation. These proteins, therefore, are not only structurally important, but in addition perform essential enzymatic processes during viral maturation [11]. Given how delicate this machinery operates, the Gag/Gag–Pol proteins have the least tolerance for incorporation of non-viral proteins.

Our goal was to insert fluorescent proteins within Gag/Gag–Pol and have these proteins packaged as part of viral Gag/Gag–Pol during assembly and viral release. A successful incorporation would then release from the confines of Gag/Gag–Pol during maturation and enter the next host cytoplasm as an individual protein.

Unlike gene delivery, proteins have much shorter biological lifespans and therefore fewer side effects. We therefore speculate that it is plausible that HIV based protein delivery systems can be used in the future.

Similar to our results, previous trials in making a fluorescent HIV focused on insertion of the fluorescent proteins in between the MA and CA domains with the fluorescent protein either permanently fused to the MA domain [26] or flanked with the natural HIV PR processing site that bridges the MA and CA domains [27] did not produce fully infectious virions. In the first case, the HIV construct was partially infectious and had a significant defect in maturation; however, the infectivity was restored when co-expressed along the parental virus [26]. In the second case a SQNY|PIV protease site was duplicated to fit in between MA and the fluorescent protein so that it could be released from the confines of Gag during maturation. To this end, we found that this construct was substantially less infectious than the wild type vector, in agreement with a recent report [30,31]. The low infectivity of MA–Fluo-CA, in addition to its defect in virion release and Gag/Gag–Pol precursor maturation, were demonstrated in our study (Figure 1B and Figure 3A). It is of note, however, that when the size of the inserted amino acid chain was small, the insertions were better tolerated. This has been demonstrated through labeling of infectious HIV virions by a biarsenical–tetracysteine system [32,33].

Our finding that the viral proteins can tolerate the insertion of a foreign protein in between NC and SP2 suggests that all Gag–Gag interactions, including MA, CA, and NC, need to be preserved for optimal assembly, budding, and protease activation. Interactions between individual Gag domains CA–CA and MA–MA as well as the structure of the immature lattice have been studied extensively [1,34,35]; our data further highlight a potential role for NC interactions during Gag assembly.

Based on the increase in size of the virions observed in EM (Figure 4), we estimate that individual virion packaging NC–Fluo-SP2 have more Gag molecules within each virion. Assuming that the Gag/surface area remain constant, and there are approximately 2000 copies of Gag within a 130 nm wild type virion, one can estimate that NC–Fluo-SP2 virions with sizes ranging from 150 nm to 200 nm will incorporate between 2600 and 4700 Gag molecules.

What are the potential uses of the developed NC–Fluo-SP2 constructs? While we developed the NC–Fluo-SP2 design primarily to study HIV budding and maturation, it is clear that the immature Gag/Gag–Pol lattice formed by the NC–Fluo-SP2 constructs has significant deviations from the wild type lattice primarily due to its enlargement (Figure 4C). While these deformed lattices are informative, their increase in size needs to be properly accounted for during analysis of any budding and maturation experiments. The reasonably high infectivity of the VSVG pseudotyped R8.2 (NC–Fluo-SP2) and its high efficiency of maturation with prominent p24 release as well as release of Fluo protein within the lumen of the virion makes these virions attractive for the study of HIV entry into cells, including capsid un-coating. These virions can also be considered as vehicles for protein delivery applications. In this context, the Fluo protein can be replaced with a protein of interest, which will be packaged during assembly along with Gag and Gag–Pol proteins. The protein of interest is then released from the confines of Gag during maturation and further releases into the cytosol of the next host during viral fusion with next host. A set of proteins within the p53 family can be considered for potential first trials [36].

## 5. Patents

A provisional patent application has been filled by the University of Utah.

## Figures and Tables

**Figure 1 viruses-11-00973-f001:**
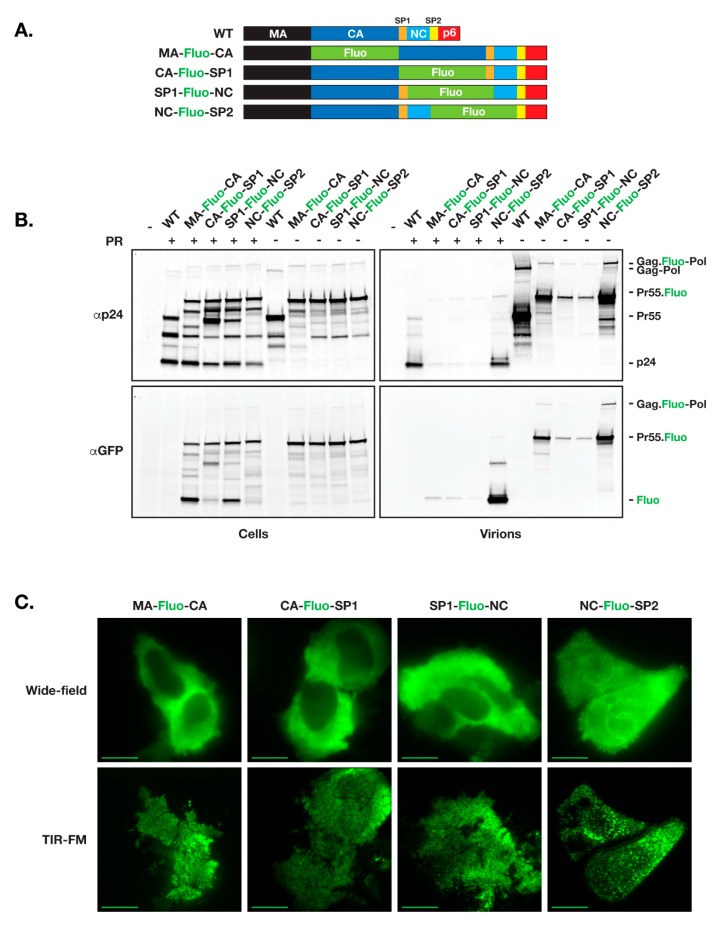
Construction and expression of HIV R8.2:Gag–Fluo vectors. (**A**) GFP-derived pHluorin (Fluo) is inserted into R8.2 between the domains of Gag as indicated with the protease cleavage sites duplicated accordingly for flanking both ends of the fluorescent protein. (**B**) Quantitative immunoblot analysis of R8.2 (WT, unlabeled) and R8.2:Gag–Fluo vectors expression using 293T cells. 24 h post-transfection, cells, and virions were collected and analyzed as described in the Materials and Methods section. Both R8.2 PR wild type (+; active protease) and PRΔD25N (−; inactive protease) were characterized. The specific primary antibodies used for immunoprobing and positions of viral proteins are indicated at left and right, respectively. (**C**) Cellular expression of R8.2:Gag–Fluo vectors was analyzed using HeLa cells. 12 h post-transfection, cells were visualized under wide-field imaging to assess cellular expression (cytoplasmic distribution), then under TIR-FM acquisition for evaluating the virions assembly on plasma membranes (dense punctae). Scale bars: 10 μm.

**Figure 2 viruses-11-00973-f002:**
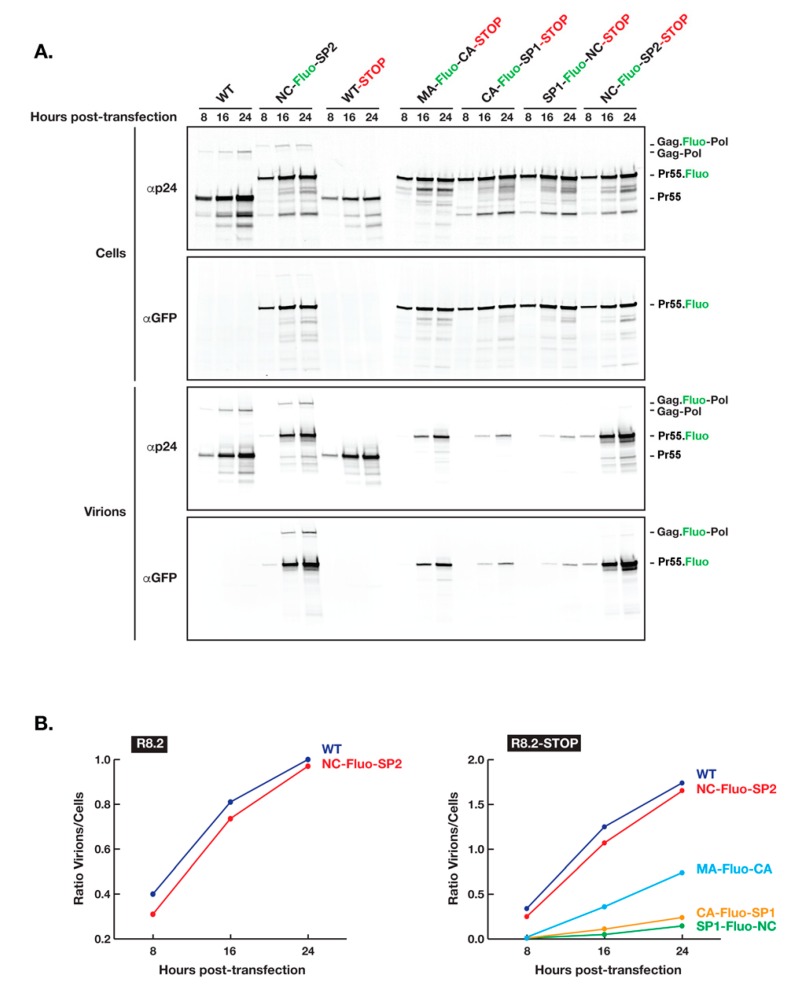
Kinetics of HIV R8.2:Gag–Fluo virion release. (**A**) The expression in 293T cells of Gag and Gag–Pol proteins are shown by R8.2 PRΔD25N and R8.2:Gag(NC–Fluo-SP2) PRΔD25N while solely Gag by R8.2–STOP variants in which a stop codon was introduced immediately after Gag p6 domain abrogating therefore Gag–Pol production. Cells and virions were collected and analyzed as described in the Materials and Methods section. The specific primary antibodies used for immunoprobing and positions of viral proteins are indicated at left and right, respectively. (**B**) Densitometry values of the panels shown in (**A**), which corresponds to the ratio of Gag in virions/cells.

**Figure 3 viruses-11-00973-f003:**
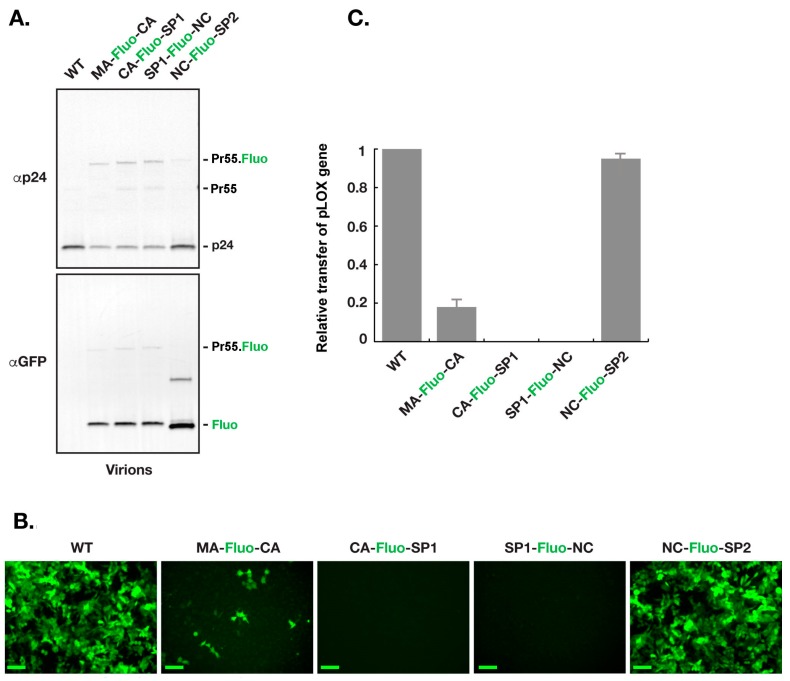
Transfer of pLOX–GFP with R8.2:Gag–Fluo virions. Virions harvested from 293T cells transfected with R8.2 or R8.2:Gag–Fluo vectors along with pLOX–GFP and pCMV–VSV–G plasmids were harvested for analysis. (**A**) Processing analysis of R8.2:Gag–Fluo virions using immunoblotting. Equal load of virions, based on density of the p24 band, were used for western blot analysis. (**B**) Equal load of virions, based on density of the p24 band, were used to infect a monolayer of Hela cells. Images show the cells 48 h post infection. Scale bars: 100 μm. (**C**) Quantitative assessment of the number of GFP expressing cells in (**B**).

**Figure 4 viruses-11-00973-f004:**
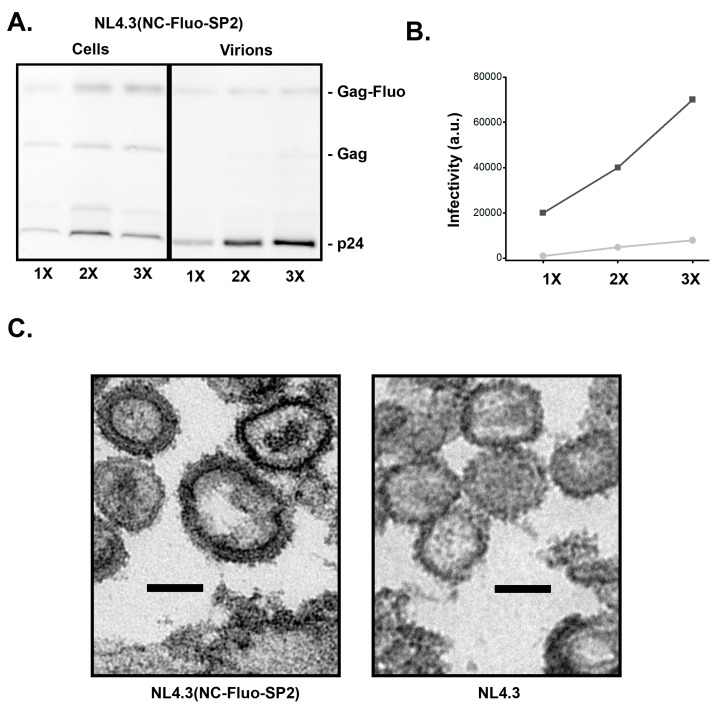
Characterization of NL4-3(NC–Fluo-SP2) virions. (**A**) Quantitative calibration of released NL4-3(NC–Fluo-SP2) virions using immunoblotting. The 1X, 2X, and 3X refer to amount of DNA used during transfection in 293 cells (**B**) Virions harvested from 293T cells transfected with NL4-3(WT) (black) or NL4-3(NC–Fluo-SP2) (gray) vectors were used to infect a monolayer of TZM-b1 cells, cells were lysed 24 h post-infection and infectivity analyzed using luciferase assay. (**C**) Electron micrograph of virions released from HeLa cells transfected with NL4-3(WT) (black) or NL4-3(NC–Fluo-SP2). Scale bar represents 100 nm.

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
