# Peer review of "Fluorescent Protein Inserts in between NC and SP2 Are Tolerated for Assembly, Release and Maturation of HIV with Limited Infectivity"

_viruses, 2019, doi:10.3390/v11110973_

Round 1
Reviewer 1 Report
In this manuscript authors reported a new construct of HIV-1 particles by inserting of fluorescent protein between the nucleocapsid (NC) and spacer peptide 2 (SP2) domains of Gag. The authors showed that the fluorescent virions were capable of budding and maturation similar to to its wild type variant. However, the infectivity was much lower in NL4-3 backbone. The fluorescent virion sizes were shown to be larger than non-fluorescent pseudovirus particles. The authors speculated that fluorescent virions have more Gag. It would be nice if the authors could elaborate how 3000 copies of fluorescent proteins were estimated in these fluorescent virion particles. Have the authors observed the similar sizes (150 to 200 nm) of virions by EM for R8.2:Gag-Fluo virions.
Author Response
Reviewer 1:
In this manuscript authors reported a new construct of HIV-1 particles by inserting of fluorescent protein between the nucleocapsid (NC) and spacer peptide 2 (SP2) domains of Gag. The authors showed that the fluorescent virions were capable of budding and maturation similar to to its wild type variant. However, the infectivity was much lower in NL4-3 backbone. The fluorescent virion sizes were shown to be larger than non-fluorescent pseudovirus particles. The authors speculated that fluorescent virions have more Gag. It would be nice if the authors could elaborate how 3000 copies of fluorescent proteins were estimated in these fluorescent virion particles. Have the authors observed the similar sizes (150 to 200 nm) of virions by EM for R8.2:Gag-Fluo virions.
Response to Reviewer 1:
We thank the reviewer for his/her thoughtful comment. We have observed a similar size increases in the R8.2:Gag-Fluo virions. The comment about the Gag number within the virion is very thoughtful. We had only provided an estimate based on the surface area change. We now provide more specifics for our estimates within the text between lines 261 and 266 as shown below:
“Finally, based on increase in size of the virions observed in EM (Figure 4), we estimate that individual virions packaging NC-Fluo-SP2 have more Gag molecules within each virion. Assuming that the Gag/surface area remain constant and there are approximately 2000 copies of Gag within a 130nm wild type virion, one can estimate that NC-Fluo-SP2 virions with sizes ranging from 150 nm-200 nm will incorporate between 2600 to 4700 Gag molecules.”
Reviewer 2 Report
The authors describe a new design for a lentiviral vector derived from HIV that encodes a fluorescent protein as part of the Gag polyprotein. The fluorescence protein is inserted between NC and SP2 and is flanked by protease sites such that it is proteolytically released during viral maturation and packaged as a solution phase marker inside the virus like particle. Potential applications of these viral particles include use as a tool for imaging HIV entry into cells. The authors show that insertion of a fluorescent protein between other viral proteins in Gag leads to a defect in particle assembly/budding and proteolytic processing. In these respects the new insertion site is superior to the established "iGFP" design (fluorescence protein flanked by protease sites between MA and CA). VSV-G pseudotyped particles are able to infect and deliver the packaged fluorescent protein to cells. Overall the paper is of substantial interest to readers interested in engineering viral particles for fluorescence labelling or as protein delivery vehicles.
The authors could consider the following minor points for improvement:
(1) Manuscript text
-Intro: The authors could cite previous approaches for tagging Gag as a tool for particle tracking in microscopy, e.g. PMIDs 18059278, 21347302
-line 109: remove “proper” from this title
-line 151: remove "dramatic"
-line 176 onwards/Figure 3/corresponding methods: This experiment is not entirely clear to me. It seems that VSVg pseudotyped viral particles containing pLOX-GFP mRNA were produced in 293T cells, collected from the supernatant and then viral titre was assessed using FACS after transduction of HeLa cells (GFP expression). The viral particles were then "calibrated" to obtain equal concentrations and analysed by Western blotting (Fig 3A) and used to infect HeLa cells (microscopy images in Fig 3B and quantification of fluorescence in Fig 3C). If this is the experimental design, then it could be more clearly communicated in the paragraph (line 176 onwards) and the figure legend. E.g. it was immediately clear to me what the authors meant by "calibrated" and what the quantification in 3C is based on. Also, if the Western in A represents virus adjusted to equal concentrations, then it might be useful to add a quantification of the bands on the blot confirming that levels are ~equal. Finally, do the authors know why the GFP band for the NC-SP2 construct is below the GFP bands of the other constructs?
-line 185: clearly state that the particles are VSVg pseudotyped to avoid confusion about what is meant by infectivity.
-line 211: “TAM" should be TZM
-line 219-222: I do not understand what is meant here, maybe rephrase this section?
-line 244: Papers before ref#31 have observed this effect, e.g. PMID 22685410
(2) Figures
-Figure 1C and 3B: add scale bar
-Figure 3A and legend: What do the authors mean by calibrated in this context? Why is there a wt Gag band for a construct that contains GFP within Gag? Presumably the blot was generated with anti-p24? Please state.
Figure 3B and legend: Was the particle concentration adjusted to equal levels prior to infection?
Author Response
Reviewer 2:
The authors describe a new design for a lentiviral vector derived from HIV that encodes a fluorescent protein as part of the Gag polyprotein. The fluorescence protein is inserted between NC and SP2 and is flanked by protease sites such that it is proteolytically released during viral maturation and packaged as a solution phase marker inside the virus like particle. Potential applications of these viral particles include use as a tool for imaging HIV entry into cells. The authors show that insertion of a fluorescent protein between other viral proteins in Gag leads to a defect in particle assembly/budding and proteolytic processing. In these respects the new insertion site is superior to the established "iGFP" design (fluorescence protein flanked by protease sites between MA and CA). VSV-G pseudotyped particles are able to infect and deliver the packaged fluorescent protein to cells. Overall the paper is of substantial interest to readers interested in engineering viral particles for fluorescence labelling or as protein delivery vehicles.
We thank the reviewer for his/hers thoughtful comments.
The authors could consider the following minor points for improvement:
(1) Manuscript text
- Intro: The authors could cite previous approaches for tagging Gag as a tool for particle tracking in microscopy, e.g. PMIDs 18059278, 21347302.
Done! Lines 252-254
- line 109: remove “proper” from this title.
Done!
- line 151: remove "dramatic".
Done!
-line 176 onwards/Figure 3/corresponding methods: This experiment is not entirely clear to me. It seems that VSVg pseudotyped viral particles containing pLOX-GFP mRNA were produced in 293T cells, collected from the supernatant and then viral titre was assessed using FACS after transduction of HeLa cells (GFP expression). The viral particles were then "calibrated" to obtain equal concentrations and analysed by Western blotting (Fig 3A) and used to infect HeLa cells (microscopy images in Fig 3B and quantification of fluorescence in Fig 3C). If this is the experimental design, then it could be more clearly communicated in the paragraph (line 176 onwards) and the figure legend. E.g. it was immediately clear to me what the authors meant by "calibrated" and what the quantification in 3C is based on. Also, if the Western in A represents virus adjusted to equal concentrations, then it might be useful to add a quantification of the bands on the blot confirming that levels are ~equal. Finally, do the authors know why the GFP band for the NC-SP2 construct is below the GFP bands of the other constructs?
Thank you for this comment, we have re written the corresponding section and legend to make the experiments clear. Please see lines 176 to 184
We don’t know with certainty why the size shift of GFP band for NC-Fluo-SP2 construct versus the other vectors occurs. However, GFP in NC-GFP-SP2 virions is fully mature after release by the HIV protease while the processing of GFP in the other constructs is very likely incomplete leading to release of GFP that remains fused to other flanking Gag domain(s) (NC, SP2 or both).
- line 185: clearly state that the particles are VSVg pseudotyped to avoid confusion about what is meant by infectivity.
Done!
- line 211: “TAM" should be TZM
Done!
- line 219-222: I do not understand what is meant here, maybe rephrase this section?
This section was shortened to:
“These lentiviral delivery systems rely on Gag and Gag-Pol proteins, which in unison drive viral assembly release and maturation.”
- line 244: Papers before ref#31 have observed this effect, e.g. PMID 22685410
New reference was inserted.
(2) Figures
- Figure 1C and 3B: add scale bar
The scale bars are now included.
- Figure 3A and legend:
What do the authors mean by calibrated in this context?
The figure legend has been clarified to reflect calibration based on p24 levels.
Why is there a wt Gag band for a construct that contains GFP within Gag? Presumably the blot was generated with anti-p24? Please state.
When immunoprobed using specific p24 antibody, HIV Gag naturally shows two bands (55, ~41 kDa, and of course p24 if processed properly by the HIV protease). The ~41 kDa is believed to be linked to a premature stop codon in the Gag sequence that is located just after the TTTTTT ribosomal slippage sequence. To this, we totally agree with the reviewer as actually the “Gag” labeled band in our previous Figure 3A-top panel may actually correspond to that truncated form of Gag-Fluo by analogy to its corresponding ~41 kDa that is produced by unlabeled Gag. To avoid any mislabeling concerning that form of Gag, we renamed the bands in the figure
- Figure 3B and legend: Was the particle concentration adjusted to equal levels prior to infection?
Yes, the legend has now been clarified.
Reviewer 3 Report
The authors propose a new GFP-based HIV vector, which has been constructed in a way that the fluorophore has been inserted in a tolerated position of the Gag-Pol polyprotein, where it can be excised by the viral protease. It is suggested that this construct may be of value for research and drug discovery.
Overall, the manuscript is clearly structured, and the data presented in logical form. Nevertheless, as the title already implies, it is not entirely clear, what the specific benefit (or even advantage over existing reporter viruses) of their construct will be. The authors omit detailing, e.g. in the discussion, for which application the construct should be used. E.g. in the context of studying Gag assembly and dynamics, it might rather be inappropriate to extend the size of the protein precursor as suggested by the EM data in Figure 4.
It would thus be necessary that the authors carefully discuss with experimental evidence, which applications will benefit from using the constructs.
Several shortcomings need attention:
As abbreviation for the clonal virus the authors use "NL4.3"; this should be replaced throughout the text with the correct term "NL4-3". In Fig.2A "Gag" is used as label on the right hand side without precising, which Gag protein is indicated. Rather use "Pr55" or "p24", etc. linie 185: "...90% as infectious" is not proper English - rephrase Fig.3A at the position of "Gag", which seems to be at the position of Pr55: - why would the size for all different Gag derivatives be the same? If the precursor also contains the size of the reporter insert, it should migrate at a different size, e.g. CA-F-SP1 vs. SP1-F-NC(?)! Fig.3C Why is this "qualitative" rather than "quantitative"? The authors used quantitative information for generating the graph. line 198 the term "transferring" is strange here (should this rather mean "transforming"?) Fig4 is mislabeled as "Fig.3"; is correct in the text Fig.4A: "Gag" should for clarity be replaced with "Gag Pr55" legend: text is missing to explain what "1x, 2x, 3x" stand for. Fig4C: What does "4.3 (Black)" stand for? should rather be 4.3 (WT) line 234: "...delivery system can be used in the future" - the authors should detail, which applications they have in mind! line 245: "Fig.1 and 3" - no such data are detailed in Fig.3 - should it rather mean Fig.4? line 248 - this sentence is flawed and should be rephrased! The discussion section contains paragraphs that simply repeat the results but do not sufficiently discuss the findings; i.e. a discussion of the findings that the infectivity is greatly reduced, effects on cleavage sequence and efficiency of the Gag precursor, information on particle integrity and stability, or information on the capability of the constructs to deliver GFP to target cells, is entirely missing! This would be an essential part for making the manuscript worth reading and for demonstrating the added value of the constructs over the state-of-the-art.
(This reviewer was unable to access the supplemented figure!)
minor points:
Throughout the manuscript, language and grammar should be carefully re-checked. Especially at multiple sites definite articles are missing, or plural and singular have been confused.
(Moreover, line 126:... the virionS assembly efficiency... change to assembly efficiency of the virions...; line 76 "on the other hand" is not correctly used here; line 123:"drastic defective yield" is not a suitable English term)
Author Response
Reviewer 3:
The authors propose a new GFP-based HIV vector, which has been constructed in a way that the fluorophore has been inserted in a tolerated position of the Gag-Pol polyprotein, where it can be excised by the viral protease. It is suggested that this construct may be of value for research and drug discovery.
Overall, the manuscript is clearly structured, and the data presented in logical form. Nevertheless, as the title already implies, it is not entirely clear, what the specific benefit (or even advantage over existing reporter viruses) of their construct will be. The authors omit detailing, e.g. in the discussion, for which application the construct should be used. E.g. in the context of studying Gag assembly and dynamics, it might rather be inappropriate to extend the size of the protein precursor as suggested by the EM data in Figure 4. It would thus be necessary that the authors carefully discuss with experimental evidence, which applications will benefit from using the constructs.
We thank the reviewer for this insightful comment and his/hers thorough evaluation of our manuscript. A paragraph has been added in the discussion covering potential uses, lines 267-280.
Several shortcomings need attention:
- As abbreviation for the clonal virus the authors use "NL4.3"; this should be replaced throughout the text with the correct term "NL4-3".
Done!
- In Fig.2A "Gag" is used as label on the right hand side without precising, which Gag protein is indicated. Rather use "Pr55" or "p24", etc.
Gag variants are now labeled accordingly.
- linie 185: "...90% as infectious" is not proper English
We apologize about this; the sentence is now properly formulated.
- rephrase Fig.3A at the position of "Gag", which seems to be at the position of Pr55: - why would the size for all different Gag derivatives be the same? If the precursor also contains the size of the reporter insert, it should migrate at a different size, e.g. CA-F-SP1 vs. SP1-F-NC(?)!
Gag labeling is now replaced in all figures by Pr55 accordingly.
The only Pr55 derivatives that are detected using p24 (CA) probing are Pr55, Pr48 and Pr41 (if produced enough to be detected) as well as mature p24. To this, all our Pr55-Fluo bands should have the same size and therefore migrate accordingly to that.
- Fig.3C Why is this "qualitative" rather than "quantitative"? The authors used quantitative information for generating the graph.
This was a mistake, we have changed qualitative to quantitative.
- line 198 the term "transferring" is strange here (should this rather mean "transforming"?)
This sentence has now been reworded.
- Fig4 is mislabeled as "Fig.3"; is correct in the text Fig.4A: "Gag" should for clarity be replaced with "Gag Pr55" legend: text is missing to explain what "1x, 2x, 3x" stand for. Fig4C: What does "4.3 (Black)" stand for? should rather be 4.3 (WT)
We agree with all these notes; it is now corrected accordingly.
- line 234: "...delivery system can be used in the future" - the authors should detail, which applications they have in mind!
This line was removed and a paragraph was introduce to discuss these issues at the end of discussion.
- line 245: "Fig.1 and 3" - no such data are detailed in Fig.3 - should it rather mean Fig.4?
We now specified in the manuscript (lines 251-252) the exact location of data (1B and 3A) that shows the defect of MA-Fluo-CA design in virion yield and infectivity when compared to the parental HIV.
- line 248 - this sentence is flawed and should be rephrased!
Done! Currently lines 256-261
- The discussion section contains paragraphs that simply repeat the results but do not sufficiently discuss the findings; i.e. a discussion of the findings that the infectivity is greatly reduced, effects on cleavage sequence and efficiency of the Gag precursor, information on particle integrity and stability, or information on the capability of the constructs to deliver GFP to target cells, is entirely missing! This would be an essential part for making the manuscript worth reading and for demonstrating the added value of the constructs over the state-of-the-art.
A paragraph was added at the end of discussion to expand on potential uses. Lines 267-280
minor points:
Throughout the manuscript, language and grammar should be carefully re-checked. Especially at multiple sites definite articles are missing, or plural and singular have been confused.
Done.
(Moreover, line 126:... the virionS assembly efficiency... change to assembly efficiency of the virions...;
We changed it to: We further assayed the virion assembly efficiency using TIR-FM imaging
line 76 "on the other hand" is not correctly used here;
Fixed.
line 123:"drastic defective yield" is not a suitable English term)
changed to: vectors showed a drastic defective yield of defect in virion release.
Round 2
Reviewer 3 Report
The manuscript has been revised, and shortcomings in the content have been corrected. The flow of text still suffers a bit from some mistakes in punctuation and missing articles in a number of sentences. (I could not correct them myself in the pdf version) --> the manuscript would benefit from a short proof-reading run for language!